# Response of *Scenedesmus quadricauda* (Chlorophyceae) to Salt Stress Considering Nutrient Enrichment and Intracellular Proline Accumulation

**DOI:** 10.3390/ijerph17103624

**Published:** 2020-05-21

**Authors:** Myung-Hwan Park, Chae-Hong Park, Yeon Bo Sim, Soon-Jin Hwang

**Affiliations:** 1The Research Institute for Natural Sciences, Hanyang University, Seoul 04763, Korea; parkmh96@konkuk.ac.kr; 2Human and Eco-Care Center, Konkuk University, Seoul 05029, Korea; qkrcoghd2@gmail.com; 3Department of Environmental Health Science, Konkuk University, Seoul 05029, Korea; sumatra8865@gmail.com

**Keywords:** N and P enrichment, eutrophication, survival, growth, green algae, N:P ratio

## Abstract

Aquatic organisms are exposed to a wide range of salinity, which could critically affect their survival and growth. However, their survival and growth response to salinity stress remain unclear. This study evaluates the growth response and intracellular proline accumulation of green algae, *Scenedesmus quadricauda*, isolated from brackish water, against dissolved salts stress with N and P enrichment. We tested a hypothesis that nutrient enrichment can relieve the dissolved salts stress of algae by accumulating intracellular proline, thereby improving survival and growth. Four levels of salinity (0, 3, 6, 12 psu) were experimentally manipulated with four levels of nutrient stoichiometry (N:P ratio = 2, 5, 10, 20) at constant N (1 mgN/L) or P levels (0.05 and 0.5 mgP/L). In each set of experiments, growth rate and intracellular proline content were measured in triplicate. The highest level of salinity inhibited the growth rate of *S. quadricauda*, regardless of the nutrient levels. However, with nutrient enrichment, the alga showed tolerance to dissolved salts, reflecting intracellular proline synthesis. Proline accumulation was most prominent at the highest salinity level, and its maximum value appeared at the highest N:P ratio (i.e., highest N level) in all salinity treatments, regardless of P levels. Therefore, the effects of P and N on algal response to salt stress differ.

## 1. Introduction

Dissolved salts are a critical factor affecting the survival and growth of aquatic biota [1,2]. In particular, freshwater and estuarine environments experience a wide range of variations in salinity, and the residing organisms have to undergo physiological adaptation to overcome dissolved salts stress. Moreover, dissolved salt stress is known to exert stronger effects on lower-level organisms [3,4], and the adverse effects of dissolved salts on algae have been documented in various studies, considering growth inhibition and death [5,6,7,8].

Biota physiologically respond to environmental stress in diverse ways [9], one of which is proline synthesis and accumulation [10,11,12]. A multifunctional amino acid proline is a signaling molecule acting as plant growth regulator by triggering cascade processes [13]. Proline is preferred as a common osmolyte in plants as regulation against stresses [11,14,15]. Various terrestrial and aquatic organisms (e.g., plants, bacteria, and protozoa) synthesize proline in response to environmental stress [2,12,16]. Algae also synthesize proline in response to various environmental stresses such as drought, UV irradiation, high temperature, exposure to heavy metals, oxidative stress, and high salinity [11]. In biological adaptation to environmental stress, proline accumulation could be a major mechanism by which organisms protect themselves from the deleterious effects of various stresses [2,11,12,17]. Furthermore, high salt level in soils and irrigation water is of major environmental concern, and salt-induced land degradation is a critical issue for food production particularly in arid and semi-arid region of the world [18]. Recent researches have been making progress in developing strategies to mitigate the salt stress effects on productivity of crop plants with due studies of the specific reaction which relates to proline accumulation of different plant species [18,19].

As the most limiting elements for algal growth, P and N are essential nutrients [20,21]; thus, eutrophication promotes the growth rate and biomass development of algae [22,23]. Several studies suggest that algal sensitivity or resistance to environmental stress is closely related to nutrients. For example, nutrients reduce dissolved salt stress on algal growth [24,25,26,27]. Dash et al. [28] also reported that nutrient enrichment in natural water significantly stimulated the growth of cyanobacteria (*Chlorococcum humicolo*) and green algae (*Scenedesmus bijugatus*) at salinity levels of 2.4 and 1.2 psu, respectively. These findings provide some insight into the response of algae to dissolved salt stress regarding the development of their populations.

However, the effect of specific environmental parameters such as nutrients (N and P) in salt stress on cellular proline accumulation of algae residing in brackish water is not well understood. Although eutrophication also stimulates algal productivity and thereby promotes bloom development, the effect of nutrient composition (i.e., stoichiometry) on algal growth response is unclear. Moreover, specific halo-sensitivity or tolerance among algal assemblages is not well understood. A model alga *Scenedesmus quadricauda* chosen for this study is ubiquitous in a broad range of freshwater systems [29], but it is often poriferous in brackish water [30,31,32], the reason fir which needs to be explored.

In this study, we hypothesize that nutrient addition ameliorates the dissolved salts stress of algae by promoting proline synthesis, which further promotes their growth rate. To test the hypothesis, we analyzed the growth rate and cellular proline content of *Scenedesmus quadricauda*, isolated from a brackish water reservoir, under various conditions of salinity and N and P concentrations.

## 2. Materials and Methods

### 2.1. Study Site and Sample Preparation

A model alga *S. quadricauda* for this study was collected from a brackish water reservoir, Gyehwa Reservoir, in Saemangeum reclaimed area, South Korea (35°46′18.5″ N, 126°38′02.1″ E). Salinity in this reservoir ranges from 1.0 to 8.6 psu with an average of 3.7 psu (practical salinity unit) during a year. The reservoir water is eutrophic based on the N and P concentrations; total nitrogen (TN) concentration ranges from 0.533–1.692 mg/L (average ± SD: 1.147 ± 0.484 mg/L) and total phosphorus (TP) concentration from 0.038–0.050 mg/L (average ± SD: 0.047 ± 0.006), during a year [33]. *S. quadricauda* is very common and often dominates algal assemblages in this brackish water body [30].

Single cells of *S. quadricauda* were isolated using a sterile Pasteur-type capillary pipette under 200–400× magnification of an inverted microscope (Axio Vert.A1, Zeiss, Germany) [34]. Isolated cells were placed in individual wells of Falcon^®^ 96 Well Cell Culture Plate (Corning, USA) with 0.2-μm-filtered (Nuclepore, Whatman, UK) and sterilized (1 h autoclaving) field water collected from Gyehwa Reservoir.

*S. quadricauda* cells were grown in a BG-11 medium [35] with a manipulated salinity of 3 psu, maintained at 25 °C with a light (14 h) and dark (10 h) cycle in a shaking incubator (V-1203P4S, Vision, Korea). The culture was illuminated at 100 μmol photons/m^2^/s using cool-white fluorescent lamps for two weeks. For the test, 10 mL of culture was transferred into a 250 mL flask with 90 mL of fresh medium.

### 2.2. Experimental Setup

Prior to the experiment, *S. quadricauda* was starved in a N and P-free BG-11 medium. The starved algal cells were inoculated into capped sterilized culture flasks containing BG-11 media with different N and P concentrations at the initial cell density of 1 × 10^5^ cells/mL. The algae were tested under the same condition as the stock culture for eight days.

To verify the effect of nutrient enrichment on the salinity stress of the alga, three combinations of N and P levels were set with four levels of salinity (Table 1). Each combination was provided with four different N or P concentrations, in which either N or P concentration was fixed to obtain different N:P ratios. The experiments were conducted in three sets under four different salinity levels (0, 3, 6, and 12 psu): (1) four P concentrations of 0.05, 0.1, 0.2, and 0.5 mgP/L with one fixed N concentration of 1 mgN/L (Exp. 1), (2) four N concentrations of 0.1, 0.25, 0.5, and 1 mgN/L with one fixed P concentration of 0.05 mgP/L (Exp. 2), and (3) four N concentrations 1, 2.5, 5, and 10 mgN/L with one fixed P concentration of 0.05 mgP/L (Exp. 3). Salinity in the medium was adjusted using NaCl (Junsei Chemical Co. Ltd., Japan). The ranges of salinity (3–12 psu) and nutrient concentration (P: 0.05 mg/L; N: 1 mg/L) for the study were selected based on the ambient condition. However, the highest salinity (12 psu) was slightly higher than natural range at the study site. Also, N and P concentrations were augmented (Exp. 3), keeping the same N:P ratios to test nutrient enrichment effect. All the experiments were conducted in triplicate.

### 2.3. Determination of Algal Growth Rate

Algal growth responses to dissolved salt treatment with nutrients adjustment were monitored at 1 d intervals for 8 days. Algal samples were analyzed after fixing with Lugol’s solution (final concentration 2%, *v/v*). One mL of Lugol-fixed sample taken from each culture flask was transferred to a gridded Sedgwick-Rafter counting chamber (Graticules S52 Sedgewick Rafter Counting Chamber, Structure Probe, Inc., West Chester, PA, USA). Cell density was measured using a Sedgwick-Rafter counting chamber under 200–400× magnification of an inverted light microscopy (Axio Vert.A1, Zeiss, Germany). Specific growth rate (per day: μ) was calculated using the following equation [36].
μ (/day) = ln(N_1_/N_0_)/(t_1_/t_0_)(1)
where, N_1_ is cell density (cells/mL) at time t_1_, and N_0_ is cell density (cells/mL) at time t_0_.

### 2.4. Proline Analysis

Proline analysis was performed at 1 d intervals for 8 days on the same day as cell density measurement. Intracellular proline concentration was determined according to the method described by Bates et al. [37]. In brief, cells suspended in 10 mL of 3% aqueous sulfosalicylic acid were centrifuged at 4000 g for 10 min to remove cell debris. To 2 mL of the supernatant, 2 mL of ninhydrin was added with 2 mL glacial acetic acid and reacted in a test tube for 1 h at 100 °C. The reaction mixture was extracted with 4 mL of toluene and mixed vigorously with a test tube stirrer for 15–20 s. Proline content was quantified spectrophotometrically at 520 nm (Model: Optizen 3220UV, Mecasys Co., Ltd., Daejeon, Korea).

### 2.5. Data Analyses

The data were expressed as the mean ± SD of triplicate experiments. All proline content and growth rate data achieved normality and variance homogeneity. Two-way analysis of variance (Scheffe and Games–Howell Post Hoc analysis) was used to test for differences in proline content and growth rate among the different levels of salinity and nutrients. The difference between samples was considered significant at *p* < 0.05. All data obtained from the different treatments were fitted to standard curves created using SigmaPlot^®^ v. 10.0 (Systat Software Inc. (SSI), San Jose, CA, USA). All statistical analyses were run using PASW^®^Statistic v. 18 (SPSS Inc., Chicago, IL, USA).

## 3. Results and Discussion

Figure 1 shows the results of Experiment 1; specific growth rates (μ) of *S. quadricauda* at different levels of salinity are plotted according to increasing P concentration with fixed N concentration (1 mgN/L). Algal growth rate increased with P concentrations (F_(4,20)_ = 12.473, *p* < 0.001) and it was evident at P concentration over 0.1 mg/L. Algal growth rate statistically significantly differed among different levels of salinity (F_(3,20)_ = 8.459, *p* = 0.003). The maximum growth rate (μ_max_: 1.59) was observed at the salinity level of 6 psu × 0.5 P × 1 N (N:P = 2), followed by 3 psu × 0.5 P × 1 N (N:P = 2) (μ_max_: 1.33), 0 psu × 0.5 P × 1 N (N:P = 2) (μ_max_: 1.08), and 12 psu × 0.5 P × 1 N (N:P = 2) (μ_max_: 0.42), while the statistical difference was shown only at the highest level (*p* < 0.05). At the highest salinity level (12 psu), the growth of *S. quadricauda* was markedly inhibited compared with the control (0 psu), while the algal growth rate was stimulated at moderate levels of salinity (3 and 6 psu). These moderate levels of salinity are similar to that of the field (annual average of 3.7 psu), indicating that the test alga might have adapted to moderate salinity levels.

Figure 2 shows the results of Experiments 2 and 3; the specific growth rates (μ) of *S. quadricauda* are plotted according to increasing N:P ratios with N manipulation. There were no significant differences of proline content among N:P ratios in different level of salinity, except for 0 psu in which proline content at higher N:P ratio (>10) significantly differed from that of lower N:P ratio (<5). The test alga clearly responded to not only P but also N concentration with the maximum growth rate (μ_max_) being statistically significantly different (*p* < 0.05) consistently under high P concentration regardless of the salinity level. The growth response of the alga to N treatment (Figure 2) was similar to that for P treatment (Figure 1). μ_max_ of *S. quadricauda* was observed at salinity levels of 3–6 psu under N treatment. Algal growth was statistically significantly inhibited (*p* < 0.05) at the highest salinity (12 psu), and the degree of growth inhibition was greater with low N and P concentrations at the same salinity level.

Proline content produced by *S. quadricauda* was statistically significantly high at the highest salinity (12 psu), compared with those at lower salinity levels (F_(3,16)_ = 5.638, *p* = 0.019). However, there were no significant differences of proline content among N:P ratios. The highest proline content (1.79 pg/cell) was observed with the addition of 0.05 mgP/L, and it decreased with increasing P concentration (Figure 3). At moderate salinity levels (3 and 6 psu), proline content increased only slightly from that of the control. These results indicate that P enrichment did not proportionally affect the proline production of *S. quadricauda*. Proline production was more evident at low P concentration under high dissolved salts stress. On the other hand, we suspect that low proline production may be compensated by high algal growth rate at high P level (Figure 2) to cope with dissolved salt stress [26]. Thus, it would be concluded that salt stress promotes proline production of *S. quadricauda* in combination with ambient P and N condition.

The effect of N on the proline synthesis of *S. quadricauda* was contrary to that of P. Proline content increased with both increasing N concentration and salinity level (Figure 4). Proline content was always high with low P concentration regardless of N concentration, consistent with the result of Figure 3. However, statistically significant difference of proline content was observed only in the salinity level higher than 6 and N:P ratio of 20. These results indicate that proline synthesis in high salt stress is likely promoted by increasing N level and N:P ratio [38].

The degree of algal tolerance to salinity stress may vary with various parameters such as light, temperature, and nutrients [26,39,40]. In this study, however, we focused on the effect of nutrients under fixed culture conditions for both light (100 μmol photons/m^2^/s) and temperature (25 °C), which were preliminarily determined to be optimal growth conditions for the test alga in the laboratory [26]. *S. quadricauda* exhibited lower dissolved salts stress (i.e., higher halotolerance) with higher concentration of nutrients under the same salinity level. Salt tolerance was less evident in the treatment without the addition of NaCl (0 psu) (Figure 2). Thus, salt tolerance and proline content of *S. quadricauda* may be affected by the composition of N and P [26]. *S. quadricauda* showed the highest proline content at the different N:P ratios; N:P ratio of 20 in 6 and 12 psu and N:P ratio 10 at 0 and 3 psu, indicating that the salinity is also a factor to affect proline synthesis. The varying proline synthesis and growth rate with N and P addition suggest that *S. quadricauda* may require differential N:P ratios for its survival and population development under different salinity levels [26,38,41].

Algae are capable of growing in some salty conditions depending on their habitats; however, their adaptability can be seriously limited above the threshold level [42,43]. In our study, *S. quadricauda* showed optimal growth at salinity levels of 3 and 6 psu, presumably because the test alga might have adapted to some saline conditions (brackish water) with salinity ranging from 1.0 to 8.6 psu (annual average 3.7 psu). Previously, Mohapatra et al. [26] reported that nutrient addition significantly increased the cell density of a freshwater green alga *Scenedesmus bijugatus* at salinity levels of 0.2–13.6 psu. Interestingly, with nutrient addition, a freshwater green alga, *S. bijugatus*, showed greater proline synthesis (10.2–16.1 pg/cell) than a brackish green alga, *S. quadricauda*, (0.06–1.79 pg/cell) in response to dissolved salts stress (Table 2). According to this comparison, freshwater algae appear to synthesize more proline than saltwater algae when exposed to dissolved salts stress. In spite of such species-specific differences in proline synthesis, the adaptive value of the difference in cellular proline accumulation remains to be clarified [11].

Proline synthesis has been reported to be an important index of stress tolerance capacity in plants and algae due to its function as a stabilizer [2,12,44,45]. Proline can act as a signaling molecule to modulate mitochondrial function, influence cell proliferation or death, and trigger specific gene expression, which can be essential for plant recovery from stress [11]. In our study, dissolved salts stress led to an increase in proline content in the cells of *S. quadricauda* (Figure 4). Cellular proline content also varied with different levels of salinity and P concentration (Figure 3). The exposure of *S. quadricauda* to increasing levels of salinity affected the growth and metabolic activity of algal cells. Algal cell growth decreased with increasing salinity from 3 psu to 12 psu under the condition of P addition (Figure 1), and proline content decreased with increasing P concentration (Figure 3). At constant N concentration, the cellular proline content of *S. quadricauda* was inversely related to P concentration. The difference was prominent at the highest salinity, where cellular proline content significantly increased at P concentration (Figure 4).

P has been reported to protect a cyanobacterium, *Anabaena doliolum*, from the toxic effects of pesticides [25]. Under dissolved salts stress, cyanobacteria attempt to adapt through Na^+^ extrusion, accumulation and synthesis of certain osmotica, and metabolic modifications [21], all of which operate at the cost of energy. Dissolved salts stress induces P deficiency in cyanobacterial cells, resulting in the utilization of phosphate granules. Rai and Sharma [21] described a strategic process of a cyanobacterium, *A. doliolum*, overcoming dissolved salts stress. Under high salinity conditions, when cellular phosphate levels become critically low, *A. doliolum* relieves P deficiency by excreting most of the enzymes and hydrolyzes extracellular P to make phosphate available for transportation into the cells [21]. In our study, the decline of cellular proline content in *S. quadricauda* with increasing P concentration may be due to the utilization of phosphate as an auxiliary nutrient source to overcome dissolved salts stress.

In contrast, cellular proline content at high salinity levels (6 and 12 psu) increased with increasing N concentration (Figure 4). Cellular proline content was statistically significantly different (*p* < 0.05) between low (N:P = 2 and 5) and high (N:P = 10 and 20) N:P ratio groups. However, the difference in proline content between the low N:P ratio groups of 2 and 5 was not statistically significant (*p* > 0.05). The proline content increased with high salinity (6 and 12 psu). At constant P concentrations (0.05 and 0.5 mgP/L), the cellular proline content of *S. quadricauda* was directly proportional to N concentration.

Proline is recognized as a source of C and N for rapid recovery from stress. It is used for stimulating growth, stabilizing membranes and macromolecules, and scavenging free radicals [17]. Rai and Abraham [46] reported that nitrate as a source of N protected the cells of a cyanobacterium, *A. doliolum*, from dissolved salts toxicity. Mohapatra and Mohanty [25] also showed that *A. doliolum* became more tolerant to pesticides when a nitrite source was added to the medium. Specifically, supplementing the medium with a N source at sublethal doses could increase the adaptability of the cells. Nitrate uptake was proportionally related to the salinity level in the medium. The uptake of sodium was minimal when nitrate was simultaneously available to the cells, indicating the interaction of nitrate with the Na^+^ carrier. Na^+^ efflux was maximal in N_2_, ammonia, urea, and nitrate in decreasing order [46]. In our study, we suspect that ammonium as a N source protected *S. quadricauda* by increasing cellular proline accumulation in response to dissolved salts stress.

Although proline synthesis is definitely correlated to dissolved salts stress [10,26,40,47,48], the synthesis of proline and other proteins is also affected by various other factors [11]. Therefore, the interaction between proline synthesis and dissolved salts stress may be complex and may reflect the combined effects of environmental factors. Moreover, cellular proline synthesis appears to vary with different levels of salinity and environment variables among different algal assemblages including green algae, cyanobacteria, and diatoms (Table 2). Further studies should include a broader range of habitats, environmental variables, and taxa in order to understand the adaptative value of the difference in cellular proline accumulation.

## 4. Conclusions

This study showed that nutrient enrichment could relieve the adverse effects of dissolved salts stress on a green alga *S. quadricauda* by promoting cell growth and proline synthesis, with differential role of N and P. Our results suggest that algal adaptability to dissolved salts stress varies with habitat conditions, in which N, P, and salinity levels affect algal survival and development in a complex manner. Also, algal sensitivity to dissolved salts stress appears to vary among different algal taxa. Our results provide deeper insight into factors underlying algal survival and development when exposed to dissolved salts stress and might be useful for further research on physiological mechanisms and differential responses of various algal assemblages, such as cyanobacteria, diatoms, and green algae, to dissolved salts stress accompanied by wastewater discharge and drought. In a real ecosystem, we must consider the potential effect of various environmental factors on the adaptative value of the difference in cellular proline accumulation.

## Figures and Tables

**Figure 1 ijerph-17-03624-f001:**
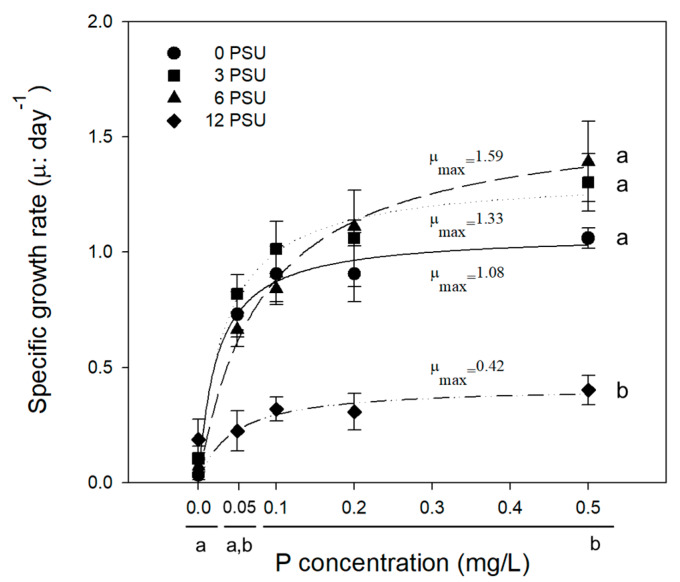
Variation in specific growth rate (μ) of *Scenedesmus quadricauda* with increasing P concentration at salinity levels of 0, 3, 6, and 12 psu. Experimental conditions are presented in Table 1 (Exp. 1). Data are presented as averages with standard deviations (bars). Lower case letters on the right side and bottom indicate significant differences (*p* < 0.05).

**Figure 2 ijerph-17-03624-f002:**
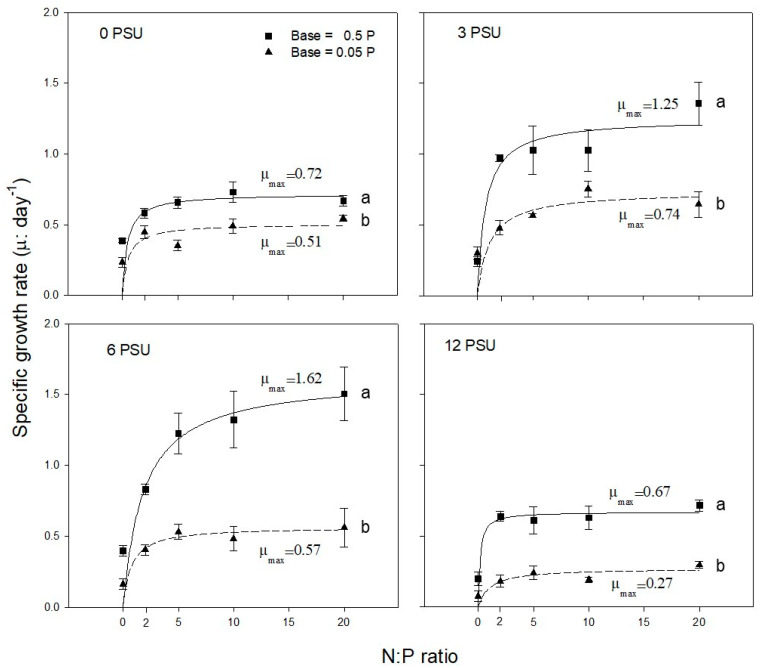
Variation in specific growth rate (μ) of *S. quadricauda* with increasing N:P ratio with two fixed P concentrations (0.05 and 0.5 mgP/L) at salinity levels of 0, 3, 6, and 12 psu. N concentration differs for the same N:P ratio because P concentration was fixed. Experimental conditions are presented in Table 1 (Exp. 2 and 3). Data are presented as averages with standard deviations (bars). Lower case letters on the right side indicate significant differences (*p* < 0.05).

**Figure 3 ijerph-17-03624-f003:**
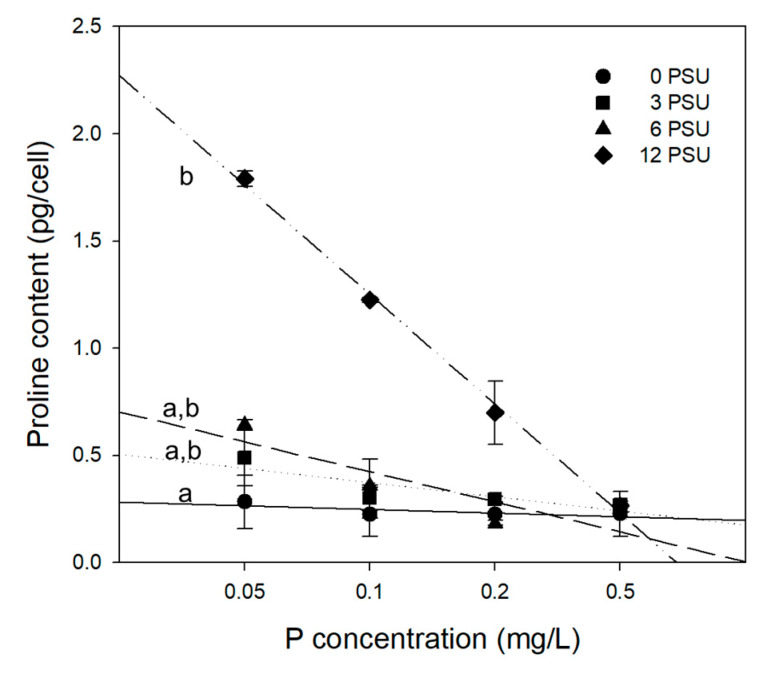
Variation in cellular proline content of *S. quadricauda* with increasing P concentrations at salinity levels of 0, 3, 6, and 12 psu. Experimental conditions are presented in Table 1 (Exp. 1). Data are presented as averages with standard deviations (bars). Lower case letters on the left side indicate significant differences (*p* < 0.05).

**Figure 4 ijerph-17-03624-f004:**
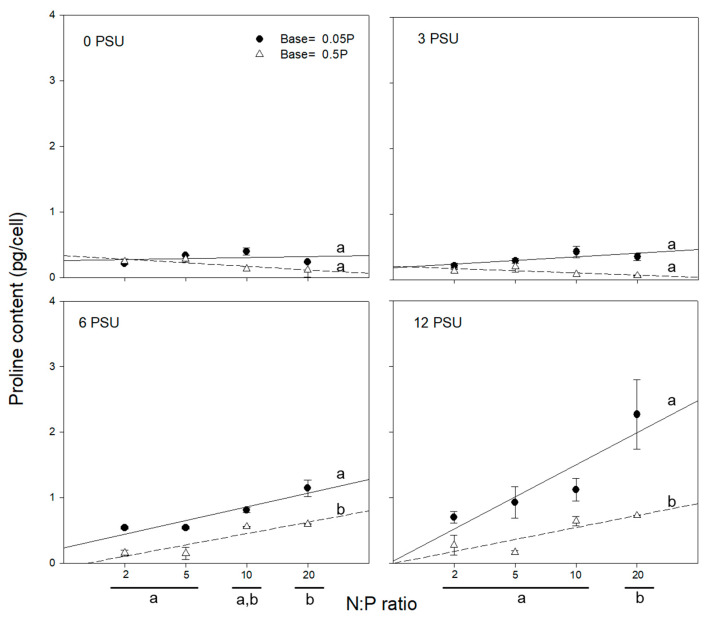
Variation in cellular proline content in *S. quadricauda* with increasing N:P ratio at fixed P concentrations (0.05 and 0.5 mgP/L) under salinity levels of 0, 3, 6, and 12 psu. N concentration differs for the same N:P ratio because P concentration was fixed. Experimental conditions are presented in Table 1 (Exp. 2 and 3). Data are presented as averages with standard deviations (bars). Lower case letters on the right side and bottom indicate significant differences (*p* < 0.05).

**Table 1 ijerph-17-03624-t001:** Experimental design of the study. Twelve treatments of N or P concentrations were administered under each salinity level.

Experimental Condition	N:P Ratio	N (mg/L)	P (mg/L)
1	2	1.0	0.5
	5		0.2
	10		0.1
	20		0.05
2	2	0.1	0.05
	5	0.25	
	10	0.5	
	20	1.0	
3	2	1.0	0.5
	5	2.5	
	10	5.0	
	20	10.0	

Notes: Salinity level applied to the experiment was 3, 6, and 12 psu including the control (0 psu). Ammonium and phosphate were used for N and P source, respectively.

**Table 2 ijerph-17-03624-t002:** Comparison of cellular proline contents in different algal species under different salinity and treatment conditions.

Class	Species	Source	Proline (pg/Cell) ^1^	Salinity (psu) ^2^	Major Treatment
Chlorophyceae	*Scenedesmus bijugatus*	[26]	10.20–16.10	0.2–13.6	Nutrient enrichment (NO_3_^−^, HPO_4_^2−^ and SiO_3_^2−^)
	*Scenedesmus incrassatulus*	[40]	0.56–0.78	10.5	Light and dark
	*Scenedesmus* *quadricauda*	This study (Exp. 1)	0.18–1.79	0–12	P addition
	*Scenedesmus* *quadricauda*	This study (Exp. 2)	0.21–2.27	0–12	Low N addition
	*Scenedesmus* *quadricauda*	This study (Exp. 3)	0.06–0.73	0–12	High N addition
Cyanophyceae	*Oscillatoria acuminata*	[47]	7.20–14.16	5–40	Different salinities
	*Spirulina platensis*	[48]	3.44 –4.64	0–12	24-epiBL addition ^3^
Bacillariophyceae	*Fragilariopsis cylindrus*	[10]	1.57–1.76	70	Hyper-salinity

^1^ Values are recalculated based on proline content per cell. ^2^ Values are recalculated for practical salinity unit (psu: permille). ^3^ 24-epibrassinolide (24-epiBL) is a substance for promoting plant growth.

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
