# Peer review of "Response of Scenedesmus quadricauda (Chlorophyceae) to Salt Stress Considering Nutrient Enrichment and Intracellular Proline Accumulation"

_ijerph, 2020, doi:10.3390/ijerph17103624_

Round 1
Reviewer 1 Report
Dear Authors,
The paper is original but due some my comments which you can see bellow can be improved.
L47-48 between the lines in Introduction part would be good to add some sentence that nowadays studied and developing strategies to mitigate the salt stress effects on productivity of crop plants due study reaction of specific reaction which is connected with proline accumulation of different plant species.
Please, see reference
Mbarki S, Sytar O, Zivcak M, Abdelly C, Cerda A, Brestic M. (2018) Anthocyanins of Coloured Wheat Genotypes in Specific Response to Salt Stress. Molecules. 23(7). pii: E1518. doi: 10.3390/molecules23071518.
Mbarki S. et al. (2018) Strategies to Mitigate the Salt Stress Effects on Photosynthetic Apparatus and Productivity of Crop Plants. In: Kumar V., Wani S., Suprasanna P., Tran LS. (eds) Salinity Responses and Tolerance in Plants, Volume 1. Springer, Cham
It would be good to add more information about Scenedesmus quadricauda like nutition qualities
Ahlgren G, Hyenstrand P, Vrede T, Karlsson E, Zetterberg S. (2000) Nutritional quality of Scenedesmus quadricauda (Chlorophyceae) grown in different nitrogen regimes and tested on Daphnia, SIL Proceedings, 1922-2010, 27:3, 1234-1238, DOI: 10.1080/03680770.1998.11901433
L108 “Proline content was quantified spectrophotometrically at 520 nm”. Please add information which spectrophotometer was used (Manufacture etc.)
It would be also good to study in the material NPK composition. Maybe authors have some data regarding it?
Th discussion part is interesting with big quantity of variants N:P ratio and salt stress effects but was studied just growth parameters and proline content. Please add information on which stage or day were collected algae species for proline analysis. I see just L83 “The algae were tested under the same condition as the stock culture for 8 days”. Please details it in proline part as well.
Author Response
Dear Reviewer,
Thank you very much for your valuable comments and suggestions, which were very helpful and constructive to improve our original manuscript. We considered them all carefully and revised the manuscript according to the reviewer comments. We modified some texts and the figures to provide better shape and clearer presentation. We hope you will find the results satisfying.
Soon-Jin Hwang, Corresponding Author

Reviewer 2 Report
Revision of the Manuscript ID: ijerph-761569 entitled “Response of Scenedesmus quadricauda (Chlorophyceae) to Salt Stress Considering Nutrient Enrichment and Intracellular Proline Accumulation”
The manuscript addresses the understanding of applying nutrient enrichment to salt stressed environments in order to improve the survival and growth rates of green algae, namely, Scenedesmus quadricauda, by augmenting proline in the algae. The topic has a potential applied interest and despite the study of proline/nutrient enrichment and salt stress is not new, the manuscript shows potential novelty in addressing this problem in Scenedesmus quadricauda, and clarifying some information about the impact of proline accumulation and nutrient enrichment in algae to resist saline stress.
However, the manuscript has small weaknesses, and I have some concerns prior accepting this manuscript to the International Journal of Environmental Research and Public Health. The authors must address the comments and or justify some putative limitations. In detail bellow.
a) Title: the title is adequate to the work and follows journal guidelines
b) Keywords: 7 keywords are used (journal standards are 3-10, so ok here); keywords are fine, they are specific to the work, but I would recommend if the editor and authors agree to replace the keywords repeated in the title, in order to expand the article search results in the future
c) Abstract: abstract is well written and addresses the different parts of the work (introduction with objective, materials and methods, results and discussion, conclusion), which I find to be a good abstract. However, it passes the word limit of the journal (200 words maximum, currently has 216 words). I would recommend to reduce the results section of the abstract (eg., you do not need to always describe the highest salinity level – 12 psu – line 24 and 27), and to reduce the conclusion remarks (line 29-30).
d) Introduction: introduction is well written, addressing the state of the art of the problem suggested by the authors. The references used support the information described. However, the affirmation provided in lines 55-57 should be rewritten, as it can lead to researchers who will read this work thinking that there is no information about proline and nutrients relations in salt stressed environments in algae. I would also like to see in the introduction more information that supports the concentrations used in the work by the authors in order to improve the methodology (namely the N and P concentrations used, as the salinity levels are explained in the materials and methods section). I would also recommend to add some more information about the algae used as a model in this work (Scenedesmus quadricauda), as I find this part of the novelty of the work, and I think it is crucial to introduce this, and not only mention it in the objective part of the introduction.
e) Abbreviations: Ok
f) Materials and methods: This section is well written and well explained, in order that other researchers can replicate the methods if the case arrives. The only thing that I would like to see complemented is the concentrations used (as I stated above in the Introduction section), in order to justify why the authors used the N and P concentrations (this can be in the Introduction section). Regarding the salinity, the authors refer in line 72 that the salinity ranges between 1.0 and 8.6, so I guess that the concentrations used in this work (line 88) are in range with those found and the 12 psu is slightly above in order to assess the effect of a higher concentration as of those found naturally, if that is the case I would like to see that explained in this section, so further researchers initiating in this area understand your concentrations.
Every subsection is well explained, and the citations used are adequate and help to complement the section.
g) Results and Discussion:
- regarding the results of the determination of algae growth rate, the authors express the results resorting to graphics that are easy to understand, but I personally found it difficult to assess the significant results, i.e., when authors state that they obtained a significantly result (e.g., line 117, 131 and 134), in the graphs I would like to see some visual information complementing the text (e.g., the use of symbols *), furthermore I would recommend the authors to further discriminate those differences (were they only P<0.05 *, or did you achieve some P<0.005 ** or P<0.001 *** for example), and also the SD should be visible in the graphs, like figure 3 and 4.
Line 116-117: add statistically before the word significantly
Line 131: change consistently high for statistically significantly different (P<0.05) consistently…
Line 134: change significantly inhibited for statistically significantly inhibited (P<0.05)…
Line 141: add statistically before the word significantly
- regarding the results of the proline content, the line 145-145 [In addition, notable difference was observed only with the lowest P concentration (0.05 mgP/L).] is repeating what was stated before, so I suggest the authors to remove it. Line 148-150: this last sentence needs further clarification, as the rapid cell growth was mostly observed indeed with higher concentrations of P in lower levels of salt stress, and these results show as the authors state, that the proline production was induced in high salt stress. The same problem stated above for the graphs of the growth rate can be seen in these ones of the proline content, except that in these ones the SD is visible. Please, further denote the statistics both in the figures and in the text if possible, as mentioned above. Lines 157-159: again, I would like a further clarification about the P concentrations and the compensation of the salt stress in the algae growth.
h) Discussion: in regards of the discussion, it is a good discussion, sufficing good information to compare, with suitable references, it is well written, and overall great, with few exceptions described bellow that the authors should take into account. Overall, in the discussion the authors try to add new information about proline content and its effect in the algae as the same with nutrient enrichment, proposing new theories that can be important in the future to understand how the algae react in saline stress.
Once the results and discussion are together, the authors should be careful to not repeat the results too much when discussing them, a revision of the discussion part should be made to minimize these repetitions.
line 168-169: insert citation to confirm the affirmation
line 172-173: author should rewrite the sentence, as the highest proline content at the N:P ratio of 20 was only observed at 6 and 12 psu (mostly at 12 psu), at 0 and 3 psu the highest proline content was observed at a N:P ratio of 10 (see figure 2). Authors should mention that, as the salinity concentration is also a factor that is taken into account in the 3 experiments.
Line 217: add statistically before significantly
Line 219: change insignificant for not statistically significant
i) Conclusion: Concluding remarks are too long, repeating information detailed in the discussion (lines 242 to 251). Needs revision, should be more concise (lines 242 to 251), and focusing more in what it could be mean for future works for example.
j) Figures and tables: overall ok, but please include the information about the statistics as referred above in detail.
k) References: references are adequate to the work, and the authors do not use self-citations, which I find a positive aspect of the manuscript. In the total of 38 citations, only 13 are from the last ten years. Even though I understand that sometimes it is not easy to find information, I think that the authors should try to include more recent citations to improve their manuscript (at least 18-20 recent citations).
In conclusion this manuscript can be accepted after minor revisions, the authors should address this smaller recommendations.
Author Response

(The authors gave the same response as above.)

Reviewer 3 Report
Thank you for providing the opportunity to review the manuscript “Response of Scenedesmus quadricauda…”. Overall, I thought this was a well-developed study and well written manuscript. I was particularly pleased with the experimental design. Bacteria growth is a function of several factors beyond N, P and dissolved salts, i.e. sunlight, other dissolved metals, temperature, etc.. So what was assessed in this work may be regarded as a bit narrow in scope and would not fully explain what would be observed in-situ, however the experimental design is complete and the results add a piece(s) to the puzzle. I only have a handful of comments and suggestions.
Line 29-30: Delete “The results proved our hypothesis”. Gives the impression this was a contest or something which of course it was not.
Line 34: Say “Dissolved salts are a critical factor…”
Line 36 an throughout paper: Replace “salt” with “dissolved salts”
Line 51: Say “nutrients, and the addition of nutrients and reduces salt stress…..”
Line 53-55: There are hundreds of studies that show nutrients enhance show cyanobacteria growth, not just Dash et at (23). This is nothing new. I suggest presenting a much more comprehensive review of nutrient impacts on cyanobacteria growth.
Line 59: Explain here what proline is.
Table 1: Explain why you did two different treatments (0.05, 0.5) for P and one for N (1.0).
Figures 1-2: What is the solid black “blob” in each figure? Is this an artifact of the graph generation? I suggest just making solids lines for each.
Author Response

(The authors gave the same response as above.)

Reviewer 4 Report
Dear authors your paper is interesting however, there are several issues that you must attend to prior publication.
The presentation of your results is difficult to follow. Graphs include a blackened area that obscures to some level results, especially in Fig. 1
Results on graphs in some cases do not seem to correspond accurately with their coordinates e.g. graph 1 at 6 psu
The different presentation of the experiments is confusing as no direct comparison could be made and the reader must try to bring them in one form or the other to understand what is the outcome of the experiments.
You should give the maxima e.g. best growth at 6 psu x 0.5 P x 1 N (N:P>2) etc.
No stastics have been shown to clarify the significant differences, especially when results are too close e.g. figure 3
Your deductionn is that proline contributes to salt stress tolerance but this has not been shown clearly. Best growth results were found at psu 3-6 where under given nutrient conditions (N:P >2) proline do not seem to differ much... You are right to say that P concentration is aiding salt tolerance but this nutrient is not associated with proline synthesis either.
Your experiments are quite useful so you must present them properly and clarify when the outcomes are really different to each other. Then you must also discuss properly your results to reflect the outcomes.
Author Response

(The authors gave the same response as above.)

Reviewer 5 Report
The manuscript entitled, Title: Response of Scenedesmus quadricauda (Chlorophyceae) to Salt Stress Considering Nutrient Enrichment and Intracellular Proline Accumulation
Journal: International Journal of Environmental Research and Public Health needs major revision before it could be considered for publication.
- Improve the Introduction. Describe the Scenedesmus quadricauda and the importance or use of proline in the cyanobacteria.
- In Table 1, the experimental code could be changed to experimental conditions 1, 2,3.
- What are inside the black figures? Any points in it?
- Possible explanations of why proline content increased or decreased with your conditions. What are the other factors that affect proline analysis? You must establish that the experiment is specific only to proline, and not other interfering impurities. Any verification or parallel experiments to verify the proline content. You must strengthen the discussion with this.
Author Response

(The authors gave the same response as above.)

Round 2
Reviewer 4 Report
Dear authors, your revised version has improved the presentation of results aa well as the discussion and conclusions. Minor issues before publication are:
In Figure 2 at 12 psu the base of 0.05 mg/L P is presented with dark circles instead triangles.
Phrase in lines 189-191 is difficult to read and must be rewritten in a way to be easy to understood.
Author Response
Response to comments (Reviewer 4)
Dear Reviewer:
Thank you very much for your comment and pointing out our mistake in Figure 2. We corrected and modified the problems according to your comments. We again appreciate very much your detailed review which was very helpful to improve both the original and revised versions of this manuscript.
Comments and Suggestions for Authors
Dear authors, your revised version has improved the presentation of results aa well as the discussion and conclusions. Minor issues before publication are:
In Figure 2 at 12 psu the base of 0.05 mg/L P is presented with dark circles instead triangles.
- Thank you very much point out this problem. We replaced this figure with new one after correcting the legend problem.
Phrase in lines 189-191 is difficult to read and must be rewritten in a way to be easy to understood.
- We modified this sentence according to your suggestion, as following (lines 191-193 in the text):
“On the other hand, we suspect that low proline production may be compensated by high algal growth rate in high P level (Figure 2) to cope with dissolved salt stress [26].”

Reviewer 5 Report
The manuscript entitled, Title: Response of Scenedesmus quadricauda (Chlorophyceae) to Salt Stress Considering Nutrient Enrichment and Intracellular Proline Accumulation; Journal: International Journal of Environmental Research and Public Health needs minor revision before it could be considered for publication.
- Improve Table 1. Delete the salinity column and make it as a Note under the table. The column information is all the same. Also, delete Exp. in Experimental conditions column. Experimental conditions should be changed to Experimental condition. Ammonium as N source and Phosphate should be deleted and put it as Notes.
- In Figures 2, 3, and 4, delete the Data presented as averages (red part). And, put these in the discussion part.
- Lines 200 to 205 and 213 to 225, Please support your claim with supporting or related literature.
Author Response
Response to comments (Reviewer 5)
Dear Reviewer:
Again, thank you very much for your comments. We corrected and modified the problems according to your comments. We appreciate very much your detailed review which was very helpful to improve both the original and revised versions of this manuscript.
Comments and Suggestions for Authors
The manuscript entitled, Title: Response of Scenedesmus quadricauda (Chlorophyceae) to Salt Stress Considering Nutrient Enrichment and Intracellular Proline Accumulation; Journal: International Journal of Environmental Research and Public Health needs minor revision before it could be considered for publication.
- Improve Table 1. Delete the salinity column and make it as a Note under the table. The column information is all the same. Also, delete Exp. in Experimental conditions column. Experimental conditions should be changed to Experimental condition. Ammonium as N source and Phosphate should be deleted and put it as Notes.
- According to your comment, we modified the Table 1.
- In Figures 2, 3, and 4, delete the Data presented as averages (red part). And, put these in the discussion part.
- According to your comment, we deleted some of figure explanation. However, we left the same description as shown in Figure 1 (“Data are presented as averages with standard deviations (bars). Lower case letters on the right side and bottom indicate significant differences (p < 0.05”) in the rest of the figures, because we believe it is necessary part in all the figures.
- Modifications are shown in lines 168-171 for Figure 2; lines 185-186 for Figure 3; and for Figure 4, we just deleted the last sentence, because the similar description already exited in line 203-204.
- Lines 200 to 205 and 213 to 225, Please support your claim with supporting or related literature.
- According to your comment, we added new related references (ref #38 and 41).
- We added a new ref. #38 to line 205, ref #26 to line 219, and a new ref. #41 with #26 and #38 to lines 224.
- We also changed the reference numbers in the rest of text after ref. 38 first cited, and in the Reference.
